# Evidence of Downregulation in Atmospheric Nitrogen-Fixation Associated with Native Hawaiian Sugarcane (*Saccharum officinarum* L.) Cultivars

**DOI:** 10.3390/plants12030605

**Published:** 2023-01-30

**Authors:** Noa Lincoln, Reinier Paul Santiago, Derek Tatum, Angel R. Del Valle-Echevarria

**Affiliations:** 1Tropical Plant and Soil Sciences Department, University of Hawai‘i, Honolulu, HI 96822, USA; 2Department of Natural Resources and Environmental Management, University of Hawai‘i, Honolulu, HI 96822, USA; 3Hawai‘i Agriculture Research Center, Waipahu, HI 96797, USA; 4Agricultural Sector Team, Deep Science Ventures, London EC3 1JP, UK

**Keywords:** agroecology, agriculture, agronomy, biogeochemistry, fixation, indigenous, nitrogen, nutrients, plants, sugarcane

## Abstract

The study of nitrogen fixation in sugarcane has a long history that has demonstrated high potential but with substantial variation in results. This 32-month study sought to assess the response of nitrogen fixation associated with sugarcane (*Saccharum officinarum* L. cvs. *‘Akoki, Honua‘ula,* and *‘Ula*) to available soil nitrogen. Plants were grown in large pots of perlite along with a fixing and a non-fixing plant control and administered liquid fertigation with varying amounts of isotopically enriched nitrogen. Assessment of nitrogen fixation utilized nitrogen isotope tracing and acetylene reduction assay in the target and control plants. Isotope enrichment and acetylene reduction assay both indicated that nitrogen fixation peaked under low nitrogen application, and declined with higher application rates, with agreement between the two methods. These results suggest that sugarcane engages in a downregulation of nitrogen fixation under high nitrogen availability, potentially explaining the high variation in published experimental results. This suggests that nitrogen management and fertilization strategy can impact the atmospheric inputs of nitrogen in sugarcane cultivation, and the potential to improve nitrogen application efficiency in cropping systems utilizing sugarcane.

## 1. Introduction

Nitrogen (N) is often the limiting factor for primary production in terrestrial ecosystems, due to its importance in fundamental biological processes and its high degree of mobility and losses in ecosystems [1]. The vast reservoir of atmospheric nitrogen gas (N_2_) is inert, and is converted to ammonium (NH_3_) through the process of nitrogen fixation, only after which N can be utilized in biological processes in various organic and inorganic forms [2]. 

In natural systems, the vast majority of N is fixed by bacteria utilizing the nitrogenase enzyme, with several types of N-fixation recognized by different groups of bacteria, such as “free-living” in soil or water, “symbiotic” bacteria associated with leguminous plants, “associative” residing in the rhizosphere of plants, and “endophytic” bacteria that dwell within organisms, with many genera able to occupy different niches. Nitrogen fixation is affected by numerous parameters including plant species and cultivar, bacterial symbionts, and environmental conditions [3]. In particular, nutrient availability has long been considered a key regulator of biological N-fixation, with nutrients such as N, phosphorus (P), and molybdenum (Mo) having strong effects on N-fixation rates. For biological N-fixation associated with plants, evidence suggests that low N-availability induces the plant-microbe association and enables this mechanism. General mechanisms include facultative or over-regulating N-fixers, which downregulate rates of N-fixation in response to increases in available soil N, or obligate fixers, which tend to demonstrate unaltered rates of fixation regardless of N availability [4,5].

Human activities now fix more nitrogen than all natural processes combined [6], and the alteration of global nitrogen cycles has multiple consequences on the health and function of ecosystems [7,8,9,10]. Although some are inadvertent, the vast majority of anthropogenic N is deliberately fixed and distributed as fertilizers to agroecosystems, primarily applying the energy-intensive Haber–Bosch method, which is used to produce nearly 300 million tons of ammonium annually. Although industrial N-fixation has resulted in the substantial increase in global food production, the dependency on human supplied N can result in negative impacts in socio-economic and environmental consequences [11].

Efforts to curtail the substantial demand in N fertilizers have included improved nutrient application methods, improved crop nutrient-use efficiencies, and the inclusion of crop-associated N-fixation. Biological N-fixation is well known to positively influence plant growth and overall health through mutualistic associations, although these relationships often become overlooked in present-day agriculture due to the use of human-mediated nutrient inputs [12,13]. The implications of facultative and obligate N-fixation strategies can play a deterministic role in plant growth and production [14], which have been explored for various ecosystems [4,15]. The behavior of facultative fixers can create a negative feedback loop in which the use of N fertilizers decreases N-fixation activity, thereby leading to an increased reliance on fertilizers.

Research into agricultural N-fixation has focused primarily on legume-related crops and their root-associated symbiosis with bacteria such as *Rhizobium*. However, more recently, multiple lines of research have demonstrated that non-legume crops can occasionally boast high levels of N-fixation, typically through associative or endophytic relationships with bacteria [16,17,18,19,20,21]. Achieving high levels of fixation in cereal crops that currently supply most of the world’s calories has been a long-cherished goal, although in general N-fixation levels are limited. The generation of plant-available nitrogen through atmospheric nitrogen fixation has tremendous potential, not only for the reduction in N inputs into large scale monocrop production, but also the utilization of such crops and their residues in diversified cropping systems to support the cultivation of other crops. 

Sugarcane (*Saccharum officinarum* L.) has a substantial history of study for N-fixation [16]. Numerous naturally occurring N-fixing bacteria have been isolated in association with and endophytic to sugarcane [16,22]. In previous studies, N-fixation in sugarcanes has ranged up to 60% [23] but has been highly variable [16], including negative results [24,25]. To our knowledge, the N-fixation strategy of sugarcane has not been investigated as to whether strategies are obligate (maintain a fixed level regardless of nitrogen resources available in the soil) or facultative (mediate the level of fixation based on available resources) fixers [4]. In this study we aimed to examine the N-fixation strategy of sugarcane as a response to N availability in the soil. We hypothesize that in response to higher levels of available soil N, sugarcane will demonstrate lower levels of atmospheric N-fixation. We formulated this hypothesis as a potential explanation to the high degree of variability and contrasting results evident in previous literature, including our own previous experimentation. The fixation strategy of sugarcane should be of high interest given the agricultural importance of sugarcane, the potential for beneficial economic and environmental impacts and the potential relevance to related cereal crops.

## 2. Results

The main treatment of the experiment involved varying levels of soil N application. The sub-treatments consisted of three sugarcane varieties and inoculation with *Gluconacetobacter diazotrophicus* PA1 5 (ATCC 49037) [26,27]. A full factorial design was implemented with these three factors: soil N availibility, sugarcane variety, inoculation. The two sub-treatments showed no significant differences between treatments; therefore, all subsequent analysis ignored sub-treatments and treated all samples as replicates to the main treatment of N level. Soil N application level significantly impacted plant biomass, total N uptake, ^15^N concentration, and ARA rates of sugarcane, as described below.

### 2.1. Sugarcane Growth

From initial seed pieces that were statistically identical, averaging 152.3 dry g, harvest dry weights in Harvests 1 and 2 resulted in significant biomass differences, with increases in biomass followed a log-linear relationship in response to increasing concentration of ammonium nitrate in the fertigation applications (Figure 1). Compared to the first harvest, second harvest growth demonstrated slightly higher biomass at lower N application and lower biomass at higher N applications; however, it followed a very similar pattern in response to nitrogen application. 

### 2.2. Nitrogen Uptake and Isotopic Concentrations

The N concentration in sugarcane demonstrated no significant differences between N treatments; however, the plants demonstrated a significant difference in N concentration over time, from a mean of 1.00% in the leaves at installation, to 0.82% at Harvest 1 sampling and 0.66% at Harvest 2 sampling. Of interest, N concentration between varieties at installation differed significantly (ANOVA, r^2^ = 0.41, *p* < 0.0001) and substantially (means of 1.04%, 0.77%, and 1.18% for ‘Akoki’, ‘Honua’ula’, and ‘Ula’, respectively), but under experimental conditions no differences were observed at the Harvest 1 and Harvest 2 sampling. 

Nitrogen isotopic ratios, which were statistically identical across treatments at installation (mean δ^15^N of 8.04‰), diverged significantly over the course of the experiment (Figure 2). For sugarcane, the response of foliar δ^15^N to the amount of N fertilizer applied was well fit by a three-parameter exponential growth model in both Harvest 1 (r^2^ = 0.96, *p* < 0.0001) and Harvest 2 (r^2^ = 0.93, *p* < 0.0001). Harvest 2 samples produced consistently and significantly lower δ^15^N values compared to Harvest 1. The two control plants, grown only during Harvest 2, displayed different response patterns compared to sugarcane and each other, with δ^15^N values that bracketed the sugarcane values at each treatment. *C. juncea* produced lower δ^15^N values that increased significantly over the two lowest N applications, followed by relatively stable values across the remaining treatments, while *B. rapa* produced higher δ^15^N values that were stable across all treatments that included N fertilization; that is, except for the N exclusion treatment. 

### 2.3. Contributions of Nitrogen from Air

Patterns of N contribution from air were evaluated both through isotopic mass balance and through the measurement of N-fixation potential through ARA. Patterns between the two approaches aligned well. 

Using total N uptake (as calculated from N concentration and dry biomass) between each time period and applying a two-end mixing model, total N contribution derived from air was determined (Figure 3). The N omission treatments demonstrated low accumulation of atmospheric N, driven by low biomass accumulation. Total inputs from atmospheric N peaked in the low N application treatments, with atmospheric N contributions declining with higher N application.

Estimation of atmospheric N-fixation by ARA suggested active N-fixation of sugarcane, primarily in the roots, and declining fixation rates with increasing N fertilizer applications (Figure 4). The growing media, *B. rapa*, and sections of sugarcane stalk all demonstrated very low rates of acetylene fixation, translating to low atmospheric N-fixation potential, that did not differ significantly across N fertilization treatments. *C. juncea*, a known obligate N-fixer, demonstrated consistently high, statistically indistinguishable, levels of acetylene reduction across all N fertilization rates. Sugarcane roots demonstrated levels of acetylene reduction that were significantly higher than the non-N-fixer control and significantly lower than the known N-fixer control. Furthermore, the potential fixation associated with sugarcane roots declined significantly in the higher N fertilization treatments. 

### 2.4. Comparison of Methodologies

The two methods used to assess N-fixation, namely Acetylene Reduction Assay (ARA) and Isotope Dilution methods, approach fixation in very different ways. ARA attempts to evaluate the potential activity of the nitrogenase enzyme, providing an instantaneous rate of N-fixation potential, while isotope dilution attempts to trace the relative inputs of soil-N into the plant. Utilizing the total estimated N fixation from ARA (sum of root and stalk measurements) and the δ^15^N values at the terminus of the experiment, the two methods were roughly fit by a log-linear relationship (Figure 5; r^2^ = 0.375, *p* < 0.001). 

## 3. Discussion

Much of the initial work on N-fixation in sugarcane occurred in low-N environments, often as a response to observations of mass-balance discrepancies in no- or low-N input cultivation systems. Productive sugarcane cultivation in Brazil, for example, was clearly uptaking much more N than was available in the soil system alone [22,28,29]. Despite the potential importance of N availability in determining N fixation levels, there have been minimal investigations examining sugarcane’s N fixation potential across nitrogen gradients. One previous study demonstrated that nitrogenase activity in sugarcane increased significantly when cutting N application in half, suggesting that sugarcane may exhibit a facultative N fixation strategy [30]. 

The results from our experiment suggest that both instantaneous (acetylene reduction) and long-term (isotopic dilution) N-fixation is affected by soil N availability. These results could explain why numerous studies, including a previous study by the lead author, have resulted in results indicating an absence of N-fixation during sugarcane growth (e.g. [24,25,31]). If examinations of N-fixation in sugarcane were conducted under high soil-N conditions, N-fixation could decline to a level that is statistically indistinguishable from a non-N-fixing control, as was seen in both the ARA and the fertilizer isotope tracing in this study. 

Although our study did not examine potential mechnisms, previous work has domenstrated that N-fixation in both free-living and symbiotic bacteria may be regulated when N availibility is increased. Studies have shown that when mineral forms of N are redily available, many N-fixing organisms will switch off N fixation [32,33], and that nitrogenase activity is inversely related to the supply of ammonium [34]. Our results could be explained by these previous findings. Under extreme deficiency, N may limit the production of nitrogenase [35], which could potentially explain the results of our N-omission treatment demonstrating low values of total N-fixation compared to treatments with low N addition. Alternatively, the plants may mediate the bacterial N-fixation by limiting carbon exchange, as has been shown to occur in the legume–rhizobia symbiosis [33,36]. 

Facultative N-fixation in agricultural crops has substantial implications for sustainable agricultural practices. The crop and management practices produce a strong feedback mechanism, whereas the more N fertilizer that is applied the less atmospheric N is sourced, increasing the need for N fertilization (or conversely, the less N fertilizer that is applied the more atmospheric N is sources, reducing the need for N fertilization). The results of this and other studies suggest that the cycle breaks down if there is no available soil N, as reduced biomass and production limits the amount of N that can be fixed. However, with relatively small additions of available N in the soil, N-fixation rates and total accumulation rapidly increase to a maximum level, before declining under high soil-N conditions. The interactions between N-fixation and fertilization were predicted as a possible explanation of previous results that demonstrate an initial peak followed by a trough in the biomass response to N-fertilization [37,38].

While the patterns of fixation appear clear in our results, several unexpected outcomes deserve some discussion. One is the lack of response to inoculation by *Gluconacetobacter diazotrophicus* PA1 5 (ATCC 49037). While this specific strain of bacteria was anticipated to increase N-fixation rates in our experiment, *G. diazotrophicus* is known to be widespread in Hawai’i and have previous been isolated from sugarcane in Hawai’i [39,40]. It seems likely that the successful inoculation would have yielded significant differences in the treatments. One potential explanation is that our source material was already inhabited by an N-fixing endophyte that resisted the inoculation of *G. diazotrophicus*, or that our attempt at inoculation was otherwise unsuccessful. Unfortunately, our controlled biological permit required we sterilize and destroy the experimental plants and we were not able to submit the plants for genetic assessment to determine what N-fixing bacteria might have been associated with our experiment. 

Additionally, no significant difference between the sugarcane varieties were observed, although upon initiation of the experiment the varieties demonstrated significantly different δ^15^N values, suggesting that there were differences of N-fixation, or at least differences of N isotope fractionation, which were occurring in the field cultivation where the material was collected. These differences quickly disappeared under the experimental conditions, although we are unclear as to why. Although our expectation was that the cultivars would perform differently in their rates of N-fixation, these were not detected in the experiment. It may be that the varieties do differ subtly in the field conditions which, over time, resulted in the divergence of the immediate soil isotope signatures and therefore the signatures of the plants in the field, but these small differences were not detectable in the controlled settings, possibly overwhelmed by the stronger lables and variations that the plants were subjected to. 

Differences between experimental Harvests 1 and 2 were also evident, with the high N treatments fixing more N from the atmosphere in Harvest 2 compared to Harvest 1. We believe that the second-harvest growth, in which the experimental plants were more root-bound in the pots, resulted in greater N losses from the pots and the behavior of the plants as if they were more N deficient than they actually were. This may also be a function of growing the control plants with the cane, which would have competed for N-uptake. 

In summary, we find that sugarcane appears to engage in substantial downregulation of atmospheric N fixation when exposed to high levels of available soil N. This was observed as high rates of atmospheric N inputs occurring in the three low-N soil availibility treatments, while atmospheric N inputs declined in the two high-N soil availibility treatments as well as the N-omission treatment. 

## 4. Materials and Methods

### Experimental Design

At the Magoon experimental greenhouse facility (21°18′24.6″ N, 157°48′36.1″ W), three Hawaiian cultivars of sugarcane (*Saccharum officinarum* L. cvs. *‘Akoki, Honua‘ula,* and *‘Ula*) [41,42] were subjected to experimental treatments to examine N-fixation under varying N-availability. Seed pieces were collected from in situ conservation plantings and surface sterilized by submerging in 10% household bleach by volume for five minutes, then rinsed throughly with clean water. Cuttings were transferred to a mist table for two weeks until rooting. One half of the seed pieces were inoculated with *Gluconacetobacter diazotrophicus* PA1 5 (ATCC 49037) by lightly misting the roots with inoculum, then starts were weighed and planted in individual 15-gallon pots of fine perlite. Random seed pieces were destructively sampled for the initial dry weight conversion and nutrient concentrations. Each pot received two types of fertilizer: a standardized amount of N-free hydroponic mixture of macro- and micro-nutrients; and varying amounts of isotopically labeled ammonium nitrate. Non-N soil nutrients were added at the rate of 1.3 g P m^−2^ y^−1^. Six levels of N were applied at the rate of 0.00, 0.38, 0.96, 1.54, 2.50, and 3.84 g N m^−2^ y^−1^, applied monthly. After 16 months, tissue samples were collected, dried, and ground for nutrient and isotopic analysis. Immediately after sampling at the 16-month time period, the cane stalks were harvested and measured for dry biomass. The samples and biomass collected at 16 months are subsequently referred to as Harvest 1. Following Harvest 1, the canes were allowed to ratoon and regrow. At 30 months each pot was planted with an N-fixing (*Crotalaria juncea*) and non-N-fixing (*Brassica rapa*) control plant. At 32 months, root and leaf samples from all canes and control plants were collected and immediately subjected to acetylene reduction assay. At the same time, additional tissue samples were collected, dried, and ground for nutrient and isotopic analyses. Immediately after sampling at the 32-month time period, the cane stalks were harvested and measured from dry biomass. The samples and biomass collected at 32 months is subsequently referred to as Harvest 2.

### 4.2. Inoculum

*Gluconacetobacter diazotrophicus* PA1 5 (ATCC 49037) [26,27] was grown following the recommendations of Tian et al (2009), with modifications. Product was obtained and is available at the American Type Culture Collection (ATCC) as catalog number 49037 (https://www.atcc.org/products/49037d-5, accessed on 25 June 2018). Briefly, *G. diazotrophicus* was grown in C2 medium for 48 hrs at 28 °C. Cells were harvested at 3500 rpm for 20 min and re-suspended in 0.8% NaCl solution to an OD600 = 0.2 for inoculation. The control solution was 0.8% NaCl in the absence of the bacterium.

### 4.3. Fertilizer Solutions

An N-free hydroponic mix was made using molecular and chelated forms of macro- and micro-nutrients. Concentrated stock solutions of each ingredient were mixed in the laboratory; a complete recipe of the fertilization solutions is reported (Appendix A). Stock solutions were added to 80 L of water and thoroughly mixed immediately prior to fertilization each month. Each pot received 500 mls of solution. An isotopically enriched N solution was made by mixing 98 atom % ^15^NH_4_^15^NO_3_ (Cambridge Isotope Labs, Tewksbury, MA, USA) with natural abundance level ammonium nitrate for a final solution of 11.85 atom percent ^15^N.

### 4.4. Acetylene Reduction

Acetylene reduction assay (ARA) was performed to assess nitrogenase activity [43]. Fresh samples were paired and incubated in either 10% acetylene gas or in air for 2 h. Gas samples were analyzed for ethylene concentrations on a Shimadzu 8A gas chromatograph (Shimadzu, Kyoto, Japan). Correlation of ARA to N-fixation used a value of 3.3 as previously determined by the lead author [25], in which samples for correlation were first incubated in 50% by volume ambient air, and 50% by volume 80% N_2_ (99% ^15^N_2_), 20% O_2_ gas for 2 h, then subsequently incubated for ARA as above.

### 4.5. Nutrient and Isotope Analysis

Nitrogen concentrations were assessed from mature leaves and a small section of the central stalk of each plant. Nitrogen isotopes were assessed from the youngest mature leaf of each plant. All tissues samples were dried at 45 °C for 48 h until a constant weight was realized, ground and analyzed for total N and ^15^N at the UC Davis Stable Isotope Laboratory using a PDZ Europa ANCA-GSL elemental analyzer coupled to a PDZ Europa 20–20 isotope ratio mass spectrometer (Sercon Ltd., Cheshire, UK) following their submission protocols (stableisotopefacility.ucdavis.edu, last accessed on 11 August 2021).

### 4.6. Calculation of N and Statistical Analysis

Total N uptake was calculated as total plant mass times the N concentration, separately for the leaves and the stalks. Isotopic mass balance was derived using a simple mixing model, using the difference in total N between starting and ending periods and applying a two end-member mixing model assuming an atmospheric ^15^N/^14^N ratio of 0.3663 [44] and applying the calculated fertilizer ^15^N/^14^N ratio of 11.85. This approach does not account for isotope fractionation rates of either atmospheric N-fixation or uptake from the soil, and therefore we focus more on the patterns between treatments rather than absolute values of calculated fixation. For all data, ANOVA and comparative means using Tukey’s HSD were used to compare the individual treatments and/or time periods. 

## 5. Conclusions

We conducted controlled experimentation using two methods to assess nitrogen (N) contributions from the atmosphere during the growth of sugarcane as a response to soil N applications. Both methods suggest moderate inputs of atmospheric N through N-fixation pathways. Furthermore, the levels of atmospheric N were affected by the amount of soil N applied, suggesting that sugarcane may downregulate N-fixtion pathways associated with its growth. Our results could explain why previous exploration of N-fixation associated with sugarcane has demonstrate in highly variable results. This work has implications for improved nutrient management strategies aimed at minimizing industrial fertilizer imputs through encouraging N-fixation. 

## Figures and Tables

**Figure 1 plants-12-00605-f001:**
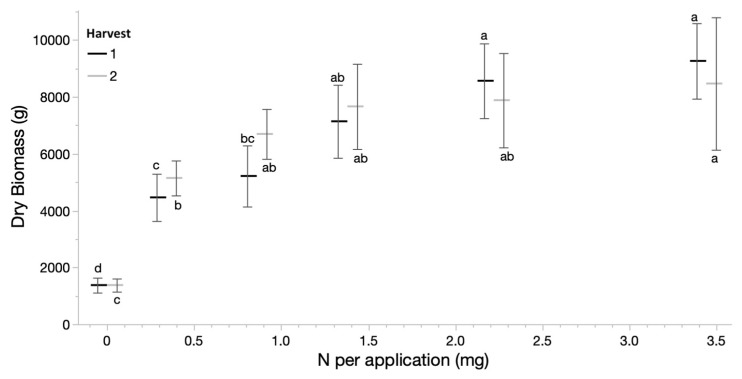
Mean dry biomass and 95% confidence intervals of sugarcane in Harvests 1 and 2 of the experiment, in response to nitrogen application per plant per fertilization period. Connecting letters denote Tukey’s HSD for each Harvest independently.

**Figure 2 plants-12-00605-f002:**
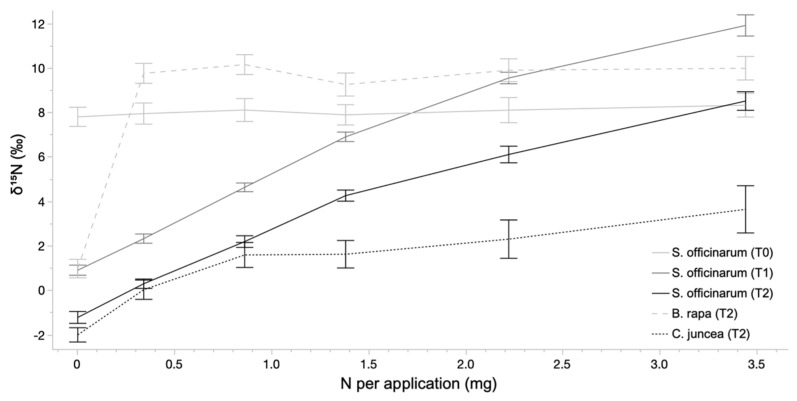
Values of δ ^15^N across nitrogen application rates per plant per fertilization period for sugarcane and control plants. Sugarcane (*S. officinarum*) is represented at initiation (T0), Harvest 1 (T1), and Harvest 2 (T2), while control plants (*B. rapa* and *C. juncea*) were only grown and measured during Harvest 2 (T2). Lines represent mean values and error bars represent the 95% confidence interval of the mean.

**Figure 3 plants-12-00605-f003:**
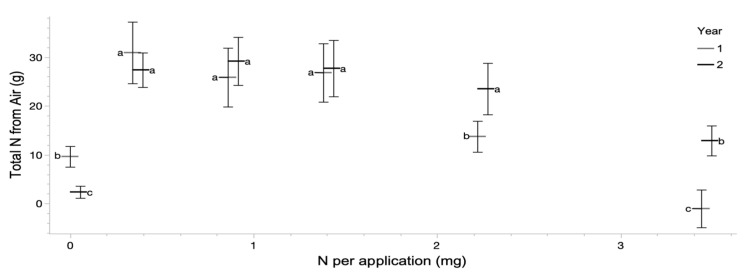
Mean and 95% confidence intervals for calculated nitrogen inputs from the atmosphere to each individual plant based on isotope mixing model and total nitrogen uptake of each replicate. Connecting letters represent Tukey’s HSD for each Harvest period independently.

**Figure 4 plants-12-00605-f004:**
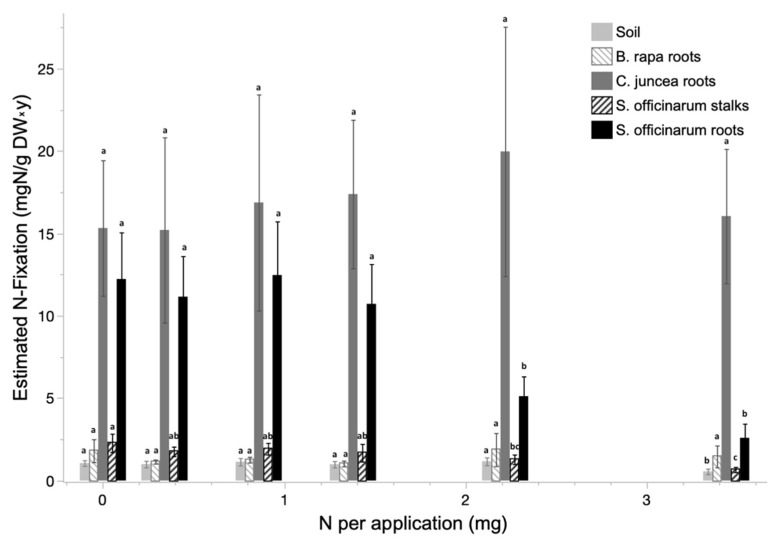
Estimated mean annual N fixation and associated standard error extrapolated from acetylene reduction assay (ARA) for sugarcane, the soilless media, and the N-fixing and non-N-fixing control plants, conducted at terminal harvest of the experiement. The connecting letters represent Tukeyʻs HSD test for each subject across treatments. The *x*-axis is the main treatment of nitrogen application per plant per fertilization.

**Figure 5 plants-12-00605-f005:**
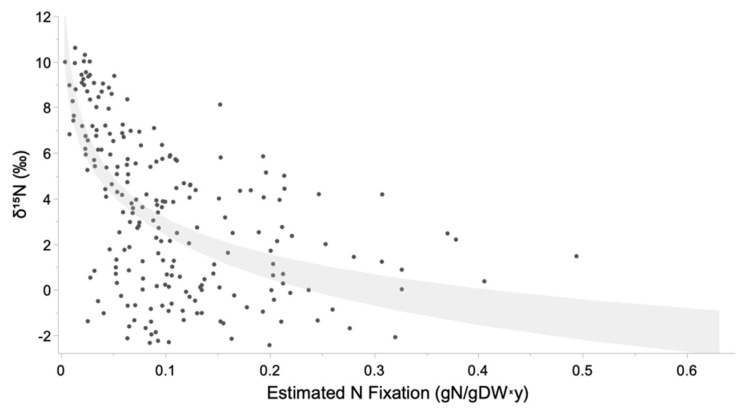
Relationship between N-fixation rates as estimated by acetylene reduction assay and the δ^15^N of plants at the terminus of the experiment, with a log-linear confidence of fit between the two methods depicted in light gray.

## Data Availability

All data are available through contacting the corresponding author.

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
