# Peer review of "Evidence of Downregulation in Atmospheric Nitrogen-Fixation Associated with Native Hawaiian Sugarcane (Saccharum officinarum L.) Cultivars"

_plants, 2023, doi:10.3390/plants12030605_

Round 1

Reviewer 1 Report

The topic of the manuscript corresponds to the journal. The study is based on a sufficient amount of experimental data. The methods of work are adequate to the task and are well described. I believe that the work can be accepted for publication after minor revision.

First of all, the hypothesis is poorly formulated. Could you suggest possible mechanisms why "response to lower levels of available soil N, sugarcane will demonstrate higher levels of atmospheric N-fixation"

I think in Results you should provide a more detailed description of the methodological aspect of the study, i.e. compare the results obtained by different methods (ARA and isotopes), highlighting this in a separate subchapter.

Also in the Results it is necessary to more clearly show the statistical differences between the indicators. Now it's not obvious.

In the Discussion, the possible mechanisms underlying the observed phenomena are poorly discussed. Could you suggest some possible mechanisms?

Finally, please make a summary. Also, this conclusion can be represented in the Abstract

Author Response

We thank the Reviewer for their useful comments and considerations that we feel aided in the improvement and clarity of the manuscript. We provide a point-by-point reply to each of the comments here:

First of all, the hypothesis is poorly formulated. Could you suggest possible mechanisms why "response to lower levels of available soil N, sugarcane will demonstrate higher levels of atmospheric N-fixation"

Our hypothesis was based on the background information out in the preceding two paragraphs, but we can see how we did not make the connections explicit. We have added the following statements in order to connect the hypothesis to the background more directly: “We formulated this hypothesis as a potential explanation to the high degree of variability and contrasting results evident in the previous literature, including our own previous experimentation.”

I think in Results you should provide a more detailed description of the methodological aspect of the study, i.e. compare the results obtained by different methods (ARA and isotopes), highlighting this in a separate subchapter.

This is a reasonable suggestion, and one we didn’t really consider. Mainly because the ARA is an instantaneous measure and the isotope dilution is a long-term accumulation, and we therefore did not really suspect very good alignment across samples without more time series ARA measurements. However, we agree it is worth presenting.  We have added a short section to this effect, including a graph that compares the final measurements of the two separate methods against each other.

Also in the Results it is necessary to more clearly show the statistical differences between the indicators. Now it's not obvious.

We apologize, but are not clear as to what the Reviewer is indicating specifically. We added clarity to the figure captions to more clearly indicate the statistical differences, and where we saw opportunity, we also attempted to clarify specifics in the text of the results.  If the Reviewer could be a little more specific or provide examples, we could better address this critique.

In the Discussion, the possible mechanisms underlying the observed phenomena are poorly discussed. Could you suggest some possible mechanisms?

We have added a paragraph to the results relating our findings to previous work and potential mechanisms in the third paragraph of the discussion.

Finally, please make a summary. Also, this conclusion can be represented in the Abstract

We have added a short summary to the end of the discussion.

Reviewer 2 Report

Dear authors,

This work studies nitrogen fixation in sugarcane as a response to N availability in the soil. The authors conclude that atmospheric nitrogen fixation in sugarcane depends on the amount of nitrogen available in the soil.

Majors:

L13 “This 32-month study” I don't understand it, then, why does it refer to year 1 and 2 in all the figures?

I do not understand very well the reason for this experimental design L217-L236. Why at 30 months cane stalks are harvested and regrow with a non-fixing plant and others in a fixing plant? Can you please better justify why you decided to do it this way?

L92: “The main treatment of N application level significantly impacted plant biomass, total N uptake, 15N concentration, and ARA rates of sugarcane, suggesting a novel finding of a facultative N-fixation mechanism associated with sugarcane”

You should start commenting on the results in a less abrupt way. First, you need to briefly explain the experimental design from which these data are derived. Without that context, it is not understood where these data come from. Where does this statement come from? “novel finding of a facultative N-fixation mechanism associated “please rewrite.

L94: “The sub-treatments”??  in relation to the above, if it does not contextualize, it is not known what this is, please clarify it.

You have to try to make it easy for the reader to understand what you say. From the way the results are displayed, this is very difficult.

L101: “years 1 and 2 resulted” What exactly does year 1 and 2 mean? I can't find it in the materials and methods, what has been going on for months? Please clarify

Figure 1. The axis of the x, mg of nitrogen, with respect to what Tons? What variety of sugarcame was this experiment done with?

Figure 2. Ideen as Fig 1. The statistical study is missing. What does T0, T1 and T2 mean?

Figure 3. grams of nitrogen in the air relative to what?

Figure 4. The statistical study is missing.  In this and in general, in all figure captions, the information is insufficient to understand it.

L167: “The results from our well-replicated experiment” It is not very good that the authors themselves openly judge this aspect. Why do you need a review then?

L192: “It seems unlikely that the successful inoculation would not have yielded significant differences in the treatments” So if the authors thought this, why did they carry out this treatment?

Minors:

L50: “Humans now fix more nitrogen” This phrase seems to indicate that humans can biologically fix nitrogen, please clarify that they are industrial processes.

L67: “fertilizers ¿?.” reference needed

Author Response

We thank the Reviewer for their extensive comments on the manuscript. We provide a point by point reply to each review comment below:

L13 “This 32-month study” I don't understand it, then, why does it refer to year 1 and 2 in all the figures?

It was an approximation and how we referred to the project data (that is, as year 1 and year 2).  We recognize that this was inappropriate and confusing to a reader.  Throughout the manuscript we have changed the description to Harvest 1 and Harvest 2, and clarified in the methods that these harvests occurred at 16 and 32 months.

I do not understand very well the reason for this experimental design L217-L236. Why at 30 months cane stalks are harvested and regrow with a non-fixing plant and others in a fixing plant? Can you please better justify why you decided to do it this way?

We were unclear in our description. Canes were not harvested at 30 months.  Canes were harvested at 16 months and 32 months.  The non-fixing controls were planted at 30 months so that their harvest could coincide with the harvest of the canes at 32 months. We can see how our wording may have caused confusion. We added a few words to the methods to hopefully clarify the timings of the experimental harvests.

L92: “The main treatment of N application level significantly impacted plant biomass, total N uptake, 15N concentration, and ARA rates of sugarcane, suggesting a novel finding of a facultative N-fixation mechanism associated with sugarcane”. You should start commenting on the results in a less abrupt way. First, you need to briefly explain the experimental design from which these data are derived. Without that context, it is not understood where these data come from. Where does this statement come from? “novel finding of a facultative N-fixation mechanism associated “please rewrite.

We have accepted this suggestion and attempted to soften and clarify the writing the first paragraph of the results.

L94: “The sub-treatments”??  in relation to the above, if it does not contextualize, it is not known what this is, please clarify it. You have to try to make it easy for the reader to understand what you say. From the way the results are displayed, this is very difficult.

We apologize for our lack of clarity and have attempted to be more clear in our writing, as above. 

L101: “years 1 and 2 resulted” What exactly does year 1 and 2 mean? I can't find it in the materials and methods, what has been going on for months? Please clarify

As per your previous comment, the terminology was changed to Harvest 1 and Harvest 2.

Figure 1. The axis of the x, mg of nitrogen, with respect to what Tons? What variety of sugarcame was this experiment done with?

The mg of nitrogen is with respect to each individual plant; that is, how much N was applied to each plant during each fertilization application.  We clarified in the caption that the mgN is “per plant per fertilization period”. The varieties of sugarcane are already clearly listed and cited in the methods, and per the first paragraph of the results, the multiple varieties did not exhibit any significant differences and the results from the three cultivars were statistically considered as a single treatment.

Figure 2. Ideen as Fig 1. The statistical study is missing. What does T0, T1 and T2 mean?

We did not feel that the connecting letters report could be easily added to this figure, nor did we feel it was as important as the different trends that are observable across the results. As such, we did not alter the figure. 

T0, T1, and T2 represent three time periods of sampling. We have added clarifications to the caption to make this clear.

Figure 3. grams of nitrogen in the air relative to what?

This is the total grams of nitrogen from the air into each plant, so it is an absolute measure (i.e. relative to each individual in the study). We have added clarity to the caption to explain this.

Figure 4. The statistical study is missing.  In this and in general, in all figure captions, the information is insufficient to understand it.

We have added the connecting letters report to the figure, and expanded the caption to help the reader understand.

L167: “The results from our well-replicated experiment” It is not very good that the authors themselves openly judge this aspect. Why do you need a review then?

We assume that the Reviewer did not like that we claimed the experiment to be well-replicated.  As such we have removed that phrase.

L192: “It seems unlikely that the successful inoculation would not have yielded significant differences in the treatments” So if the authors thought this, why did they carry out this treatment?

We believe the reviewer may have mis-interpreted this statement due to our use of a double-negative in our language. We DID expect differences IF inoculation had been successful.  We were surprised that there were no differences in our treatments that were inoculated. We carried out this treatment exactly because we expected increased fixation with inoculation, as seen in other studied.  Which is why we go on to suggest that it is possible that our inoculations were unsuccessful. We removed the double negative in this sentence hopefully to improve clarity.

Minors:

L50: “Humans now fix more nitrogen” This phrase seems to indicate that humans can biologically fix nitrogen, please clarify that they are industrial processes.

We changed to state "human activities"

L67: “fertilizers ¿?.” reference needed

This is a logic statement. As such, we do not believe that a citation is necessary.

Reviewer 3 Report

The manuscript presents a systematic study on the N-fixation biology of sugarcane. Various experiments and methods were used to assess when the sugar cane fixes nitrogen. These data are useful for both the sugar cane industry as well as for fundamental research. The manuscript is clear and contains only minor, accidental linguistic errors (e.g.  line 13, sough should probably be sought, line 222, an "of" is probably missing). 

Line 221: It may appear to be a standard procedure, yet I would appreciate if you described how the seeds were surface sterilized, since there are after all minor modifications of this procedure.

General comment, as fine as the approach is with regard to focusing on the achievements of Gluconacetobacter and possibley other not yet identified N-fixing species in sugar cane, the accompanying, non-fixing microbiome may also affect the performance of the N-fixing species. I simply suggest that you give this a thought and address this, if rational enough to bring it up, in your discussion part. 

Author Response

We thank the reviewer for the positive review of the manuscript, and provide a point-by-point reply to each comment here:

The manuscript presents a systematic study on the N-fixation biology of sugarcane. Various experiments and methods were used to assess when the sugar cane fixes nitrogen. These data are useful for both the sugar cane industry as well as for fundamental research. The manuscript is clear and contains only minor, accidental linguistic errors (e.g.  line 13, sough should probably be sought, line 222, an "of" is probably missing). 

We have made the small typographical corrections that you pinted out, and a couple more throughout the manuscript.  Thank you.

Line 221: It may appear to be a standard procedure, yet I would appreciate if you described how the seeds were surface sterilized, since there are after all minor modifications of this procedure.

Very true.  Our method was described as “submerging cuttings in 10% household bleach by volume for 5 minutes, then rinsing thoroughly with clean water.”

General comment, as fine as the approach is with regard to focusing on the achievements of Gluconacetobacter and possibley other not yet identified N-fixing species in sugar cane, the accompanying, non-fixing microbiome may also affect the performance of the N-fixing species. I simply suggest that you give this a thought and address this, if rational enough to bring it up, in your discussion part. 

We appreciate this comments. It is a perspective we are very interested in even if not the purpose of this paper. We have tried to acknowledge this in the discussion, but could not find a place to add it without it feeling non-sequitur. 

Reviewer 4 Report

Manuscript is well organized and in the form presented is acceptable for publication.

Only few Minor Revisions

Line 95 (line 223), for bacterium Gluconacetobacter diazotrophicus the strain is Pal5 or Pa1 5?

For Gluconacetobacter diazotrophicus Pal5 need to give the citation as

Bertalan, M., Albano, R., de Pádua, V. et al. Complete genome sequence of the sugarcane nitrogen-fixing endophyte Gluconacetobacter diazotrophicus Pal5. BMC Genomics 10, 450 (2009). https://doi.org/10.1186/1471-2164-10-450

If it is Pa1 5 we need to know the difference between stains 49037 and 49038 ((Gluconacetobacter diazotrophicus (Gillis et al.) Yamada et al.)

Further for Gluconacetobacter diazotrophicus PA1 5 (ATCC 49037) we need a Reference.

According to https://bacdive.dsmz.de/strain/40 the Sequence accession description is Gluconacetobacter diazotrophicus PA1 5 strain PAl 5 strain PAl 5; ATCC 49037 (is it correct?)

Please specify the product e.g. is available at as ATCC® Catalog No. 49037™ (https://www.atcc.org/products/49037d-5)

Author Response

We thank the reviewer for the positive consideration of our manuscript.  The reviewer is correct that this was ATCC PA1 5 (49037).  We have added the citations and availability statements, including the link, to the manuscript. 

Round 2

Reviewer 2 Report

Dear authors,

I believe that the authors have responded favorably to most of my suggestions and I accept the corrected version of the paper.